# Colistin Update on Its Mechanism of Action and Resistance, Present and Future Challenges

**DOI:** 10.3390/microorganisms8111716

**Published:** 2020-11-02

**Authors:** Ferdinando F. Andrade, Daniela Silva, Acácio Rodrigues, Cidália Pina-Vaz

**Affiliations:** 1Division of Microbiology, Department of Pathology, Faculty of Medicine, University of Porto, 4200-319 Porto, Portugal; agr@med.up.pt (A.R.); cpinavaz@med.up.pt (C.P.-V.); 2Farmanimal Veterinary Centre, 2500-323 Caldas da Rainha, Portugal; 3Clinical Microbiology Department, Porto University Hospital Center, 4099-001 Porto, Portugal; daniela.sfsilva@gmail.com; 4Abel Salazar Institute of Biomedical Sciences, University of Porto, 4050-313 Porto, Portugal; 5CINTESIS—Center for Health Technology and Services Research, Faculty of Medicine, University of Porto, 4200-319 Porto, Portugal; 6Burn Unit, S. João University Hospital Center, 4200-319 Porto, Portugal

**Keywords:** colistin, polymyxin, antimicrobial resistance, multidrug-resistant gram-negative organisms

## Abstract

Colistin has been extensively used since the middle of the last century in animals, particularly in swine, for the control of enteric infections. Colistin is presently considered the last line of defense against human infections caused by multidrug-resistant Gram-negative organisms such as carbapenemase-producer *Enterobacterales*, *Acinetobacter baumanni*, and *Pseudomonas aeruginosa*. Transferable bacterial resistance like mcr-genes was reported in isolates from both humans and animals. Researchers actively seek strategies to reduce colistin resistance. The definition of guidelines for colistin therapy in veterinary and human medicine is thus crucial. The ban of colistin use in swine as a growth promoter and for prophylactic purposes, and the implementation of sustainable measures in farm animals for the prevention of infections, would help to avoid resistance and should be encouraged. Colistin resistance in the human–animal–environment interface stresses the relevance of the One Health approach to achieve its effective control. Such measures should be addressed in a cooperative way, with efforts from multiple disciplines and with consensus among doctors, veterinary surgeons, and environment professionals. A revision of the mechanism of colistin action, resistance, animal and human use, as well as colistin susceptibility evaluation is debated here.

## 1. Introduction

Colistin (Polymyxin E) is a polycationic peptide antimicrobial, discovered in 1949 in Japan, produced by *Bacillus polymyxa*. It belongs to the polymyxin class of antibiotics, with hydrophilic and lipophilic properties. The polymyxin group includes five different chemical compounds (polymyxins A, B, C, D, and E) [1,2]; only two polymyxins are used clinically: polymyxin B and colistin, also called polymyxin E [3]. For clinical use, two forms of colistin are available: a prodrug, colistin methanesulfonate sodium (CMS), for parenteral use, and colistin sulfate (CS) for oral, inhalator, or topical use [4].

It was first used in human and veterinary medicine in 1952, but between the 1970s and 1980s, its medical use was almost abolished, thereby persisting the veterinary use. In recent years, colistin has begun to be used in humans as a last resource agent in the case of infections by multidrug-resistant Gram-negative bacilli, notably by carbapenemase-producing *Enterobacterales*, *Pseudomonas aeruginosa*, and *Acinetobacter baumannii* [5,6,7,8]. The World Health Organization (WHO) and other government agencies such as Health Canada have reclassified colistin in the category of “very high importance for Human Medicine” [9,10].

Colistin is considered by the European Medicines Agency (EMA) as antibiotics critically important in human medicine—Category B—“Restrict”. This means its administration in veterinary medicine should be limited to reduce the danger to public health. Quinolones (fluoroquinolones and other quinolones), cephalosporins of either third and fourth generation (excluding those with beta-lactamase inhibitors), and polymyxins are drugs included in this category. Antibiotics in Category B should be reserved for treatment when antimicrobials in Categories C or D are not, from a clinical point of view, effective and there is no other alternative. In general, antibiotic utilization should rely upon the performance of an antimicrobial susceptibility test (AST), particularly those included in Category B [11].

## 2. Mechanism of Action

Colistin L-diaminobutyric acid, positively charged, binds through electrostatic interaction to the negatively charged phosphate groups of lipid A, an important component of lipopolysaccharide (LPS) of Gram-negative bacilli [12]. The lipid A plays a crucial role in bacterial permeability and exchange with cell exterior [6]. Colistin displaces the divalent cations of calcium (Ca^2+^) and magnesium (Mg^2+^) in a competitively way, impairing the LPS three-dimensional structure. Colistin then inserts its hydrophobic terminal acyl fat chain, causing an expansion of the external outer membrane (OM) monolayer. A permeabilization of OM occurs, allowing colistin to get through OM as a self-promotion. The synergy exhibited by colistin and other antimicrobials with hydrophilic properties such as b-lactamics, gentamicin, rifampicin, meropenem, and tigecycline is explained by this process [13]. The phospholipid bilayer of the inner membrane (IM), present only in Gram-negatives, loses its stability, because of the action of colistin by including hydrophilic groups in the fatty acid chains, changing its integrity and leading to its destruction, failing to maintain cellular content, leading to cell lysis [6]. Binding to lipid A, colistin also exerts an anti-endotoxin activity [1], preventing the induction of shock by endotoxin. Colistin essentially solubilizes the bacterial cell membrane, resulting in a bactericidal effect.

## 3. Colistin Resistance 

Resistance to colistin can be explained by a variety of mechanisms. Until 2015, it was thought to be acquired solely via chromosomal point mutations. As LPS is the target of colistin, any change in it can change the behavior of colistin [7]. Salmonella and E. coli are able to modify LPS by changing lipid A through the biosynthesis of 4-amino-4-deoxy-L-arabinose (L-Ara4N) and/or phosphoethanolamine (PEtn). Its biosynthesis is associated with chromosomal mediated resistance, dependent on two-component response regulators and sensor kinase systems: PmrA/PmrB and PhoP/PhoQ [14,15,16]. The first system PmrA/PmrB also controls the pmr HIJKLM operon, which promotes the synthesis of N4-aminoarabinosis, which in turn, when bonding chemically the fractions of lipid A, changes the negative charge of the cell membrane by neutralizing the negatively charged phospholipids. This resistance mechanism is exhibited by Pseudomonas aeruginosa [17]. The resistance to colistin expressed by Acinetobacter baumannii is based on the suspension of LPS production. This absence of LPS production may result from the inactivation of a lipid A biosynthesis gene, lpxA, lpxC, or lpxD, and leads to resistance to colistin due to the absence of lipid A [18].

Regarding the resistance expressed by *K. pneumoniae* to polymyxins, its resistance mechanism is based on the inactivation of the mgrB gene and, in deregulation, the two-component regulatory systems PhoPQ and PmrAB. Inactivations in the mgrB gene were responsible for the exhibition of resistance by K. pneumonia to colistin. When the PhoP/PhoQ signaling system is activated, a small transmembrane regulatory protein, composed of 47 amino acids, is produced: the MgrB. This protein prevents the phosphorylation of PhoP, probably by suppressing the PhoQ kinase or promoting the phosphatase. As PhoP phosphorylation is found to increase the transcription of the mgrB gene and, on the other hand, MgrB inhibits this phosphorylation, then this protein is clearly a mediator of negative feedback in the PhoQ/PhoP signaling circuit [19,20].

Initially reported in China in 2015 [21], and later in Asia, Africa, Europe, and America [22,23,24,25], a plasmid mediated colistin resistance, *mcr-1* gene, was described in *Escherichia coli*. The *mcr*-1 is an enzyme that changes lipid A present in LPS with a metabolite of phosphoethanolamine, suppressing its binding to colistin.

The chances of transferring bacterial resistance to colistin along with resistance to broad-spectrum cephalosporin are significantly raised by the fact that a single plasmid can co-harbor mcr-1 together with the extended-spectrum beta-lactamase gene (ESBL). This poses huge challenges when treating infections with Gram-negative etiology. A mcr-1 gene located chromosomally was detected in two colistin-resistant *E. coli* isolates, collected from calves [26].

Later, in Belgium, a new colistin resistance gene, *mcr-2*, was also discovered, carried by a plasmid in *E. coli* isolates from porcine and bovine biologic samples and also co-harboring ESBL genes [27].

Interestingly, the coexistence of several mcr genes on *E. coli* does not necessarily mean a significant difference in terms of minimum inhibitory concentration (MIC), compared with resistant isolates of *Salmonella* only carrying the single plasmid *mcr-1* gene [28]. The *mcr-1* gene found in resistant *Enterobacterale* isolated from swine was often associated to a low level of resistance; MICs of 4 or 8 mg /L were reported for most isolates, which are only 2–4 times higher than the European Committee on Antimicrobial Susceptibility Testing (EUCAST) clinical breakpoint (2 mg/L) [21,28,29] or Clinical Laboratory Standard Institute (CLSI). Strains with minimal inhibitory concentrations inferior to 2 mg/L are considered susceptible by EUCAST protocol and intermediate by CLSI; susceptible category was recently eliminated by CLSI.

Colistin resistant bacteria also share resistance to other types of antibiotics used such as aminoglycosids; tetracycline; sulfonamide and trimethoprim; lincosamide; b-lactamics; quinolones; and third generation cephalosporins involving different mechanisms of resistance such as enzymatic, efflux, impermeability, or point mutations [29,30,31,32,33].

Meanwhile, seven additional *mcr* homologues (*mcr-3* to *mcr-9*) have been identified in *Enterobacterales* [34,35] and PCR tests have been developed to enable their detection [36]. This mechanism of resistance can be acquired during therapy and easily transmitted, increasing the spread of resistance.

The emergence of resistance to colistin, one of the few remaining therapeutical alternatives for patients infected with *K. pneumoniae* resistant to carbapenems and other important antimicrobial groups, is thus a major concern, notably among humans.

## 4. Epidemiology of Colistin-Resistance 

According to the European Centre for Disease Prevention and Control (ECDC), in 2017, isolates resistant to colistin represented 8.5% (2.4% of all reported *K. pneumonia* isolates and only sporadically on *E coli*). Greece and Italy were the countries responsible for the vast majority (88.5%) of these reports [37].

On the other hand, the same data source reported in 2016 that only 51.3% of all *P. aeruginosa* isolates were colistin susceptible. Concerning *Acinetobacter* spp., colistin susceptibility data reach up to 51.3% of all isolates [37].

However, the ECDC in 2018 repeated the warnings that these findings may not be representative for Europe as a whole and should be interpreted with caution because of the low number of isolates tested, the relatively high proportion of isolates from areas of high resistance, and the technical complexities involving colistin susceptibility tests [38]. The possibility of transmission of colistin resistant *E. coli* between species is possible, particularly from swine [39] or from pets [40] with close relationships with humans.

## 5. Colistin Susceptibility Testing Assays

For both individual and epidemiological purposes, it is essential to have accurate antimicrobial susceptibility testing (AST) regarding bacteria isolated from infected humans and from animals. Because colistin binds to several laboratory materials, many technical issues are raised and the results of colistin susceptibility can be tricky and often incorrect. A survey in 2017, among laboratories providing data, revealed that the majority of the laboratories did not test colistin susceptibility locally or used methods that are not recommended [38].

The Clinical & Laboratory Standards Institute (CLSI) and the European Committee on Antimicrobial Susceptibility Testing (EUCAST), American and European organizations respectively, responsible for the standardization of AST laboratory protocols, joined efforts to produce colistin susceptibility rules. They have issued recommendations confirming that microdilution is, for the time being, the only susceptibility testing method valid for colistin.

Several difficulties related to the methodology arise in the assessment of susceptibility to colistin. For performing an AST, the 20776-1 standard method of microdilution (BMD) is the reference method and the only one now validated for *Enterobacterales*, *P. aeruginosa*, and *Acinetobacter* spp. Culture media, colistin formulation, and even the kind of plastic used on microplates impact and were standardized. However, such a method is cumbersome and requires a long time to achieve a result (minimum of 24–48 h). Susceptibility testing by other methods, including agar dilution, disk diffusion, gradient diffusion, and automated methods (such as Vitek2, Phoenix), is not recommended, making most of the available epidemiological data not accurate. Newer methods including molecular approaches still need optimization; while they can detect the few, known resistance genes, they cannot formally be considered a susceptibility assay. The detection of resistance genes, such as *mcr*, means that the strain is resistant to colistin, although its absence does not necessarily mean susceptibility. A novel method based upon flow cytometry that allows AST determination in a maximum of 2 h versus 2 days, directly from positive blood cultures or from colonies (2 h versus 1 day in that case), can change the diagnosis paradigm [41,42]. Quick AST reports by the microbiological lab, in an era of an increase of antimicrobial resistance, are thus urgently needed [43].

## 6. Colistin: Human Use

In recent years, with the scarcity of antimicrobials options available, there is a growing interest in the deprecated antibiotic, the colistin. Colistin has established itself as a last resort therapy in infections of gram-negative multidrug-resistant (MDR) etiology. Because of the nephrotoxicity and neurotoxicity exhibited, colistin was abandoned in the early 1980s, but its resurgence as a resource therapy in critically ill patients is taking place. Regarding its intravenous use, we can identify three milestones in the course of colistin: 1950–1970, for gram-negative infection in general; 1990–2000, in cystic fibrosis with gram-negative MDR infections; and after the passing of the millennium until present time, against infection of gram-negative MDR etiology.

International consensus recommendations for colistin therapy provide a guide for its optimal clinical use [44]. It is often the last line of defense against multidrug-resistant Gram-negative bacteria, namely, *Enterobacterales*-carbapenemase producers [45,46], *Pseudomonas* spp., and *Acinetobacter* spp. Despite being Gram-negative, pathogens such as *Brucella* sp., *Burkholderia cepacia*, *Helicobacter pylori*, *Edwarsiella* sp., *Moraxella catarrhalis*, *Neisseria* sp., *Proteus*, *Providencia*, *Serratia*, and *Stenotrophomonas mallei* do not generally show susceptibility to colistin and, with regard to *Campylobacter* sp., its susceptibility has a wide range of variation. However, it is particularly used in critical clinical conditions such as bacteremia or sepsis and pneumonia associated with mechanical ventilation (VAP) in the intensive care unit (ICU). For other several clinical conditions, such as for urinary tract infections, meningitis, osteomyelitis and joint infections, infections of the gastrointestinal tract, pneumonia, abscess, pyoderma and/or soft tissues infections, and eye and ear infections, colistin is seen as an alternative treatment. On the other hand, myasthenia gravis and hypersensitivity to polymyxin are conditions in which the use of colistin is contraindicated and there are no scientific data to support or establish safety in its use in pregnant and lactating women. Colistin should be administered carefully with dose correction accordingly and tight surveillance in patients with renal impairment because of its nefrotoxicity. Intravenous polymyxins have been evaluated for the treatment of serious multidrug-resistant *P. aeruginosa*, *Acinetobacter baumannii*, and *Enterobacterales* infections [1].

Intravesical administration of colistin in therapies of persistent urinary tract infections with *Acinetobacter baumannii* etiology [47] has been described, as well as intrathecal administration in the case of meningitis because of the low permeability to colistin by the hemato-encefalic barrier [48].

Selective decontamination of the digestive tract (SDD) is a practice used in intensive care, in which colistin can be used orally along with a non-prolonged treatment of broad-spectrum parenteral antimicrobials. Despite the existence of scientific data supporting this intensive care practice to reduce bloodstream infections and mortality, when it extends over time, the rising of resistance to colistin among ESBL-producing *K. pneumoniae* isolates has been seen [49,50,51]. Recent studies with contradictory conclusions show that this practice is far from the consensus [52,53].

Some authors propose inhaled colistin as monotherapy or associated with systemic therapy in the treatment of pneumonic processes or in chronic pulmonary infection of multi-resistant Gram-negative etiology [54].

Pharmacodynamics (PD) and pharmacokinetics (PK) bring emerging data on colistin as monotherapy. It is very difficult to entrust that monotherapy will produce enoughplasma levels, with the real possibility and danger of promoting the resistance. For this reason, colistin combination therapy seems to increase clinical success and reduce resistance emergence. In vitro experimental data also support the preference for a combination therapy, particularly updated studies through the usage of dynamic models. Several studies, both in vitro and in vivo (mouse model), have pointed out that the association of colistin with other antibiotics, such as rifampicin and imipenem, has a better performance compared with colistin alone in the case of multidrug-resistant Gram-negative bacteria [55,56]. However, Lagerbäck et al. recommended that such an approach should be further studied in vivo, considering that colistin combination therapy should be reserved for critical ill patients with multidrug-resistant Gram-negative bacteria pneumonia [56]. This therapeutic approach still needs additional clinical studies, addressing its real effectiveness and, not less important, its resistance potential impact.

Lee et al. conducted a study reporting that the combined therapies with colistin and carbapenem, tigecycline, or rifampicin were more effectively in comparison with the colistin monotherapy treatment regarding positive carbapenemase *Klebsiella pneumoniae* [57]. The difficulty lies in defining clinical protocols supported by pharmacodynamics and toxicodynamics for combined colistin therapies that allow the highest possible bacterial lethality with the lowest chance of toxicity, and simultaneously prevent the emergence of resistance.

The new cephalosporins associated with beta-lactamases inhibitors like ceftazidime/avibactam could be an alternative for some carbapenemase producing bacteria (especially *Enterobacterales* KPC and OXA-48), and clinicians are left in a challenging situation: either they keep treating with old well-known drugs or they adopt new antibiotics with insufficient evidence [58].

Regarding *Pseudomonas*, a synergetic activity of colistin with ceftazidime has been reported as well as the combination of colistin with rifampicin and amikacin. The combination of colistin with rifampicin has also provided a synergetic antibactericidal effect for infections caused by *Pseudomonas aeruginosa* MDR [1].

Both in vitro and in vivo studies are clear to corroborate that antimicrobial resistance expands when bacterial agents are exposed to inferiorly unsuitable colistin gradients, observed in a significant proportion of patients [59], particularly in isolated heteroresistant strains such as, for example, those of Acinetobacter baumannii [60].

The association with carbapenem, doripenem, or meropenem, as well as with several other antimicrobials, such as rifampicin, glycopeptides, daptomycin, and fusidic acid, generally ineffective in Gram-negatives, revealed a synergistic effect against *Acinetobacter baumannii*. The effective mechanism of such unlikely combined therapies rests on the polymyxin effect, changing the outer membrane permeability, allowing or/and making easier, despite being large molecules, the entry of antimicrobials into the cytoplasm [46].

Discrepancies now can be found on the antimicrobial susceptibility report; recently, CLSI (2000) abolished the classification of susceptible for all gram-negative bacteria *Enterobacterales*, *Pseudomonas*, and *Acinetobacter*, only naming as intermediate or resistant, although EUCAST (2000) still maintains the susceptible category.

Therapeutic drug monitoring of colistin is warranted because its pharmacokinetics is quite variable and because its therapeutic window is narrow [61]. This aspect makes the management of the colistin treatment challenging, stressing the need for the availability of practical and simple methods with a short response time, including a drug monitoring assay.

## 7. Colistin: Veterinary Use and Its Impact

At present, colistin is an antibiotic still widely used in veterinary medicine, mostly in pigs, for the oral treatment of intestinal infections caused by *Enterobacterales* [62]. The administration of antibiotics like colistin in animals supported the growing of modern farm animal productions, allowing successful weaning, higher animal density, and probably higher viability of economic control of pathologies caused by *E. coli* infections such as that caused by verotoxigenic *E. coli* (VTEC) [22]. Colistin is poorly absorbed through the gastrointestinal tract. This fact stresses the emergence of colistin resistance as a result of selective pressure upon the intestinal microbiota [22]. Pigs treated with colistin had overall higher proportions of resistant isolates compared with pigs not treated [62]. Colistin is also used orally in calves for the treatment of gastrointestinal diseases caused by Gram-negative bacteria. This use may also explain the isolation of colistin-resistant bacteria from calves, despite the lack of solid data about the treatment of these animals with colistin [32].

Notably, the most common use of colistin in swine industrial production worldwide is the oral route and for prophylatic purposes [63]. Colistin was mainly administered via feed, but also through drinking water [64]. This practice involves medicating all animals in the same farm, resulting in treating animals with clinical symptoms together with healthy ones. The European Medicines Agency (EMA) made a categorization for colistin use, which was reviewed in 2016 [64]. Colistin therapies with prophylactic purposes carry a high risk in the emergence of resistance, and thus should be banned. Its clinical use should be limited to enteric infections caused by susceptible non-invasive *E. coli* and supported by an AST whenever possible [43,64].

Although colistin, in oral treatment, is generally used as monotherapy [22], there are some pharmaceutical forms on the market with colistin, under the sulfate salt chemical form, that allow combined therapy. The most common association with other antimicrobials is with β-lactamics and mainly amoxicillin [65]. In vitro data show the combination of amoxicillin with colistin reveals a synergistic or additive effect against pathogenic *E. coli* of avian source [66]. Combinations of colistin and amoxicillin plus zinc oxide (ZnO) during the pre-weaning and weaning periods incorporated in the swine feed were also used [67]. A study carried out in fecal samples of weaned piglets showed that the combination of three antimicrobials, colistin, bacitracin-zinc, and chlortetracycline, inhibits the rise in tet resistance genes (tetX, tetC, tetL, and tetW) [68]. Despite the suppression in tet resistance genes, no data exist regarding resistance to colistin after using this combined therapy. Because of a lack of evidence, the combined therapy cannot be scientifically sustained, particularly among pigs. Considering all the available updated knowledge, the Committee for Medicinal Products for Veterinary Use (CVMP), basing the decision on the precautionary principle, recommended the Market Authorization withdrawal for all Market Authorization Holders and all veterinary pharmaceutical formulas associating colistin and other antibiotics [64].

The use of colistin as a growth promoter of farm animals, still adopted specially in Asia, should be banned. Low sub-inhibitory concentrations of antibiotics used to improve animal growth have been linked to the emergence of antibiotic resistance [22]. Notably, antimicrobials for animal growth promotion can generally be purchased without veterinary control even in the European Union. High sanitation, by controlling the microbial burden in the farm, is a key factor to help to ban antimicrobial misuse.

As a general attitude with all antimicrobials, but in particular with colistin, the veterinary surgeon should ensure that the antimicrobial prescribed is applied strictly for the treatment of sick animals as recommended in compliance with label instructions. Any deviations from the EMA guidelines should be justified and recorded [11,64]. In this context, extra-label use of colistin such as, for example, in countries where this antibiotic is not approved in swine, must take place within a valid veterinary justification relying upon AST reports and without the availability of other therapeutic alternatives.

Further research is mandatory in order to determine the withdrawal period allowing the risk reduction of animals sent to slaughter harboring colistin resistant bacteria carrying *mcr* genes.

Animal-to-human transmission of *mcr-1* resistance raises issues concerning the consequences of using colistin in veterinary medicine (in pet treatment, farm animal’s production, and its penetration in the consumption food chain) in terms of human health [39]. *mcr-1* might be present in the environment and transmissible by several routes to humans. Therefore, it could also add to the potential transferring of the *mcr-1* gene from animal to human. Such transfer routes require additional and more comprehensive studies.

The veterinary surgeon should seek that colistin use only targets clinical disease and consider banning its use whenever feasible. Educational and awareness campaigns for pig farmers and employees are essential. Records detailing colistin use in animals are equally critical as they provide support to allow the development of policies and guidelines and to assess the effect of possible interventions. It has been shown that colistin-resistant *E. coli* was isolated from healthy individuals without prior medical colistin use [69].

Additionally, reports from Italy, the second country using colistin adjusted for biomass under exposure in European Union [64], describe the detection of colistin-resistant *E. coli* from wild rabbits (*Oryctolagus cuniculus*) and wild hares (*Lepus europaeus*) that were not previously treated with colistin [70]. Consequently, wildlife may also play a role in colistin resistance, being a potential reservoir in the environment, and contribute to its transmission to other animals and/or humans through contaminated food and water or by direct human and animal contact.

Apart from the previous isolation of bacteria resistant to colistin from manure, water, migratory birds, and vegetables [23,71,72,73], the ecotoxicity of this antimicrobial and its impact is an issue to take into consideration [74,75]. The colistin, at high levels, present in swine farms’ wastewater, is related to the toxicity exhibited in the bacteria responsible for oxidizing ammonia [74]. The biotransformation of xenobiotic substances along with the chemical transformation of ammonia into nitrites in wastewater treatment plants is ensured by these bacteria responsible for the oxidation of ammonia [74]. The significant damage inflicted on the intestinal epithelium and the greater expression of stress-related genes in the earthworm *Eisenia fetida* are other rising ecotoxic markers revealing the environmental toxicity caused by the elimination of colistin in wastewater [75].

It should be remembered that administered antibiotics are not 100% metabolized by animals or humans; many are excreted in an active form via the feces or urine. This fact is even more marked with oral administration of colistin, which is very poorly absorbed by the gastrointestinal tract [22].

Composting methods have been reported to be able to eliminate on average 50–70% of some antimicrobials, such as chlortetracycline, monensin, and tylosin [76], also helping to reduce the relative amounts of the *bla*_TEM_, *sul3*, and *erm*(B) genes in manure [77]. Unfortunately, they do not work in reducing the amounts of colistin or *mcr* genes in pig manure. The method in which manure is handled or treated has a variable impact on the effectiveness of reducing antibiotic resistance genes (ARGs) in pig manure; that is, aerobic biofiltration of manure seems to be more effective to reduce *erm*(X) than other ARGs such as *erm*(F), *erm*(B), and *tet*(G), while mesophilic anaerobic digestion and lagoon storage reduced none of these ARGs [78]. A great deal of controversy still involves the effectiveness of these biological treatments of manure, such as lagoons and composting, in ARG reduction [76]. To ensure the destruction of resistant bacteria and ARGs present in animal and human waste is mandatory in further research. It is necessary to ensure the effectiveness of waste treatment processes, particularly with regard to pig manure. The agricultural role of swine manure in soil fertilization and the absence of regulations and guidelines should be considered to control and monitor the use of swine manure [22].

## 8. Conclusions

An interdisciplinary collaboration and communication between all the specialists involved is necessary to develop a universal strategy that fosters the reduction of the emergence of resistance to colistin in human, veterinary medicine, and the environment. The scientific community, experts, and government authorities urge for a reduction of colistin usage, recommending its prescription only for the treatment of infections as a last resource under strict circumstances. Colistin use should involve a scientific evaluation of the effectiveness of other drugs and implement alternative strategies against infections caused by *Enterobacterales* whenever possible, avoiding antimicrobials at all, particularly in preventive strategies. The *mcr-1* gene was isolated on four continents and, most worrying, from sources other than food animals, such as the environment as well as from human isolates. In addition, some *E. coli* isolates harboring the plasmid-encoded *mcr* genes, now known up to *mcr*-9, were also ESBL or carbapenemase enzyme producers. All such facts cause great concern about the eventual loss of colistin effectiveness regarding the treatment of multidrug-resistant Gram-negative bacilli in humans. Epidemiological data could not be accurate owing to technical issues involved in the susceptibility evaluation. Rapid and accurate ASTs are urgently needed. This highlights the need for the judicious use of colistin in order to avoid the development of pan-resistant strains that are not bound by land borders and do not require a passport or visa.

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
