# Peer review of "Colistin Update on Its Mechanism of Action and Resistance, Present and Future Challenges"

_microorganisms, 2020, doi:10.3390/microorganisms8111716_

Round 1

Reviewer 1 Report

This article provides a great overview of colistin with diverse perspectives in molecular biology, epidemiology, veterinary/clinical medicine, and environmental science. The authors delivered each subject and pointed out controversial results in a logical manner. With future challenges, the authors stressed the need for collaboration with interdisciplinary teams. I would suggest a minor revision to clarify several points in this paper.

Abstract

  1. Page 1, line 21: In my opinion, information about transmissible ESBL genes is not necessary for the abstract regarding colistin.

Text

  1. Page 1, line 62: Please clarify that L-diaminobutyric acid “of colistin” interacts with lipid A of Gram-negative rods.
  2. Page 1, line 75: Please clarify that the action on “lipid A” is limited to Gram-negative rods.
  3. Page 1, line 92: Please briefly explain the colistin resistance mechanisms in Klebsiella spp.
  4. Page 3, line 110-112: The authors may clarify what point should be delivered from this sentence.
  5. Page 3, line 116: The authors may clarify the meaning of “two mechanisms”. Does it mean two resistance mechanisms of mcr-1 and mcr-2 genes or mcr genes and chromosomal mutations?
  6. Page 3, line 134-135: As far as I know, genetic mutations of two-component regulatory systems in Gram-negative bacteria isolated from humans are mainly attributed to colistin resistance rather than transmissible mcr genes.
  7. Page 3, line 141: Please provide further information on which spp. were tested in the ECDC available data.
  8. Page 6, line 251: Please provide the reference.
  9. Page 7, line 327-328: The authors may limit this conclusion to the use of colistin, not of all antimicrobials.

Author Response

I appreciate the criticisms of our article that I think when answered make it richer and more interesting. I hope that the answers presented will satisfy your opinion on this manuscript.

Abstract

  • Page 1, line 21: In my opinion, information about transmissible ESBL genes is not necessary for the abstract regarding colistin.

Answer:

Removed from the text

Text

  • Page 1, line 62: Please clarify that L-diaminobutyric acid “of colistin” interacts with lipid A of Gram-negative rods.

Answer:

Lines 56-58 – I hope that this way leaves no room for doubt.

  • Page 1, line 75: Please clarify that the action on “lipid A” is limited to Gram-negative rods.

Answer:

Lines 56-58 – I hope that this way leaves no room for doubt.

  • Page 1, line 92: Please briefly explain the colistin resistance mechanisms in Klebsiella spp.

Answer:

Lines 90-99 -  Klebsiella's resistance mechanisms are described in these lines. New references have been introduced.

  • Page 3, line 110-112: The authors may clarify what point should be delivered from this sentence.

Answer:

Lines 111-112 - The consulted and referred author recommends this action to avoid the spread of antimicrobial resistance.

  • Page 3, line 116: The authors may clarify the meaning of “two mechanisms”. Does it mean two resistance mechanisms of mcr-1 and mcr-2 genes or mcr genes and chromosomal mutations?

Answer:

Lines 116-118 I hope that when rephrasing the sentence there is no room for doubt.

Reviewer 2 Report

Dear Authors,

I am glad having the opportunity of reviewing the Manuscript 935664 entitled ‘Colistin update on its mechanism of action and resistance, present and future challenges’ for ‘Microorganisms’.

I believe major alterations and improvements are needed before the paper is suitable for publication.

Altogether, reading the paper there is the feeling that authors did not reviewed each aspect in the same depth. For example ‘Colistin resistance’ is quite well analytically written, and ‘Human use’ is not. Text about ‘Veterinary use and its impact’ is 3-4 times more extended combined to human use and its impact. In a review about present and future challenges I would like to read more about the relationship between different resistance mechanisms and virulence of isolates, clinical impact of colistin-resistance to (intensive care unit) patients and more about alternative antimicrobial regimens currently used for treating severe infections due to colistin-resistant strains.

Some other comments:

Introduction: 

  • ‘The risk to public health resulting from veterinary use needs to be mitigated by specific restrictions’: Please omit this phase. 

Colistin resistance

  • ‘ Animal-to-human transmission of mcr-1 resistance, raises issues concerning … Therefore, it could also ... Such transfer … studies… The detection of isolates … isolation’: Please omit this text or transfer text to another section as it is not strictly relevant to colistin-resistance mechanisms. 
  • ‘Colistin resistant bacteria share resistance also to other types of antibiotics also used such as aminoglycosids, tetracycline, sulfonamide and trimethoprim, lincosamide, b-lactamics, quinolones, and third generation cephalosporins’: Please make a statement about the common resistance mechanisms.
  • ‘Plasmid-borne mar genes… spread globally’: Please omit this phase. 

Resistance of colistin epidemiology’: rephrase to ‘Epidemiology of colistin-resistance’.

Colistin susceptibility testing assays

  • ‘A survey in 2017, among providing data laboratories revealed that the majority of the laboratories did not test colistin susceptibility locally or used methods that are not recommended’: Please transfer text to the end of the previous paragraph

Colistin: Human use

  • As you read the text, you have the feeling that the sentences are not quite good logically connected one to another.  

Author Response

I appreciate the criticisms of our article that I think when answered make it richer and more interesting. I hope that the answers presented will satisfy your opinion on this manuscript.

Altogether, reading the paper there is the feeling that authors did not reviewed each aspect in the same depth. For example ‘Colistin resistance’ is quite well analytically written, and ‘Human use’ is not. Text about ‘Veterinary use and its impact’ is 3-4 times more extended combined to human use and its impact. In a review about present and future challenges I would like to read more about the relationship between different resistance mechanisms and virulence of isolates, clinical impact of colistin-resistance to (intensive care unit) patients and more about alternative antimicrobial regimens currently used for treating severe infections due to colistin-resistant strains.

Answer:

The text on colistin in Veterinary Medicine is more extensive because a part related to the environment has also been included. In any case, we increased the text on the use of colistin in humans as recommended. I hope we have met your recommendations.

Some other comments:

Introduction:

‘The risk to public health resulting from veterinary use needs to be mitigated by specificrestrictions’: Please omit this phase.

Answer:

Removed from the text.

Colistin resistance

‘ Animal-to-human transmission of mcr-1 resistance, raises issues concerning … Therefore, itcould also ... Such transfer … studies… The detection of isolates … isolation’: Please omit this text or transfer text to another section as it is not strictly relevant to colistin-resistance mechanisms.  

Answer:

Transfered to Veterinary use and its impact - line 293.

‘Colistin resistant bacteria share resistance also to other types of antibiotics also used such as aminoglycosids, tetracycline, sulfonamide and trimethoprim, lincosamide, b-lactamics, quinolones, and third generation cephalosporins’: Please make a statement about the common resistance mechanisms.

Answer:

Lines 126-130 - The mechanisms of action of the other antimicrobials are mentioned as recommended, although in a very succinct way so as not to lose the main objective: colistin, nor to make the article too long.

‘Plasmid-borne mar genes… spread globally’: Please omit this phase.

Answer:

Removed from the text.

Resistance of colistin epidemiology’: rephrase to ‘Epidemiology of colistin-resistance’.

Answer:

line 138 Rephrased as recommended.

Colistin susceptibility testing assays

‘A survey in 2017, among providing data laboratories revealed that the majority of the laboratories did not test colistin susceptibility locally or used methods that are not recommended’: Please transfer text to the end of the previous paragraph

Answer:

Lines 155-158 Transferred as recommended.

Colistin: Human use

As you read the text, you have the feeling that the sentences are not quite good logically connected one to another.

Answer:

I sincerely hope that after the recommended changes the text will be clearer, more logical and more interesting.

Round 2

Reviewer 2 Report

Dear Authors, 

I am glad having the opportunity of reviewing your revised Manuscript 935664 entitled ‘Colistin update on its mechanism of action and resistance, present and future challenges’ for ‘Microorganisms’. 

It is clear that you made a sincere effort to improve the quality of the Manuscript. The text is greatly improved, but I still believe that a second revision is required before Manuscript is potentially suitable for publication in ‘Microorganisms’. Please work more with the text about human use of colistin and provide us more information about clinical aspects of colistin-resistant Acinetobacter baumanni, right as you did with Klepsiella and Pseudomonas.

Some other comments: 

Colistin: Human Use

  • ‘Contraindications for colistin are patients with myasthenia gravis and hypersensitivity to polymyxin’: please change to ‘Contraindications for colistin are myasthenia gravis and hypersensitivity to polymyxin’
  • ‘Intravenous polymyxins… and meningitis’: There is a repetition regarding previous sentence. 
  • ‘intratecal’ please change to ‘intrathecal’ and ‘colistin intravenous’: please change to ‘intravenous colistin’

Colistin: Veterinary use and its impact 

  • Please try to combine sentences and avoid throughout the text the repetition of  ‘the veterinary surgeon should ensure…’
  • Please transfer ‘Epidemiological studies have described the possible horizontal transmission of colistin resistant E. coli from swine (Olaitan et al., 2015) or from companion animals (Zhang, 2016) to humans following close contact’ to other more relevant part of the text. 
  • Please tranfer ’The use of colistin for prophylaxis purposes in swine is also directly related to the high amounts of colistin in waste water and its presence in the environment (Rhouma et al., 2016) to other more relevant part of the text. 

Conclusion

  • Please complete the sentence: ’This highlights the need of the judicious use of colistin in order to avoid the development of pan-resistant strains,..’

Author Response

First of all, we thank you for your interest and contribution that allowed the article to become more interesting. We hope that this time we have satisfied your expectations and removed all doubts that the manuscript has raised. However, we appreciated your critical sense, which I understood as constructive and for that reason improved the article. We present the answers, point by point, in the hope that your opinion on the improvement of the article may coincide with ours and you could consider it ready for publication.

 “Please work more with the text about human use of colistin and provide us more information about clinical aspects of colistin-resistant Acinetobacter baumanni, right as you did with Klebsiella and Pseudomonas

Answer:

We increase the text about human use and new references were added.. An International Consensus provides a guide for colistin therapy in human therapy was included - Regarding Acinetobacter baumannii we have included more information Lines183 to 191, 222 to 228, 239 to 241 and  251 to 264.

Some other comments:

Colistin: Human Use

Contraindications for colistin are patients with myasthenia gravis and hypersensitivity to polymyxin’: please change to ‘Contraindications for colistin are myasthenia gravis and hypersensitivity to polymyxin’

Answer:

Line 197 - Rephrased as recommended.

Intravenous polymyxins… and  meningitis’: There is a repetition regarding previous sentence.

Answer:

Lines 201 to 204 - Rephrased as recommended.

‘intratecal’ please change to ‘intrathecal’ and ‘colistin intravenous’: please change to‘intravenous colistin’

Answer:

Lines 208 and 209 - Changed as recommended.

Colistin: Veterinary use and its impact

Please try to combine sentences and avoid throughout the text the repetition of ‘the

veterinary surgeon should ensure…’

Answer:

Line 314 and 328 - Rephrased as recommended.

Please transfer ‘Epidemiological studies have described the possible horizontal transmission of colistin resistant E. coli from swine (Olaitan et al., 2015) or from companion animals (Zhang, 2016) to humans following close contact’ to other more relevant part of the text.

Answer:

Transferred to lines 150 to 152 (Epidemiology of colistin-resistance).

Please transfer ’The use of colistin for prophylaxis purposes in swine is also directly related to the high amounts of colistin in waste water and its presence in the environment (Rhouma et al., 2016) to other more relevant part of the text.

Answer:

Transferred to lines 369 and 370.

This highlights the need of the judicious use of colistin in order to avoid the development of pan-resistant strains.

Answer:

Sentence finished accordingly. Lines 386 and 387.
